# Phenotypic and Genomic Characterization of *Streptomyces pakalii* sp. nov., a Novel Species with Anti-Biofilm and Anti-Quorum Sensing Activity in ESKAPE Bacteria

**DOI:** 10.3390/microorganisms11102551

**Published:** 2023-10-13

**Authors:** Michelle Chávez-Hernández, Jossue Ortiz-Álvarez, Jesús Morales-Jiménez, Lourdes Villa-Tanaca, César Hernández-Rodríguez

**Affiliations:** 1Departamento de Microbiología, Escuela Nacional de Ciencias Biológicas, Instituto Politécnico Nacional, Prol. de Carpio y Plan de Ayala, Col. Sto. Tomás s/n, Ciudad de México 11340, Mexico; michel_ale_09@hotmail.com (M.C.-H.); lourdesvillatanaka@gmail.com (L.V.-T.); 2Programa “Investigadoras e Investigadores por México”. Consejo Nacional de Humanidades, Ciencias y Tecnologías (CONAHCYT). Av. de los Insurgentes Sur 1582, Crédito Constructor, Benito Juárez, Ciudad de México 03940, Mexico; jossue.ortiz@conahcyt.mx; 3Departamento el Hombre y su Ambiente, Universidad Autónoma Metropolitana-Xochimilco, Calzada del Hueso 1100, Villa Quietud, Coyoacán, Ciudad de México 04960, Mexico; jimorales@correo.xoc.uam.mx

**Keywords:** *Streptomyces*, biofilm formation inhibitors, quorum-sensing inhibitors, ESKAPE bacteria, *Serratia marcescens*, prodigiosin inhibition

## Abstract

The increasing number of infections caused by antimicrobial multi-resistant microorganisms has led to the search for new microorganisms capable of producing novel antibiotics. This work proposes *Streptomyces pakalii* sp. nov. as a new member of the Streptomycetaceae family. The strain ENCB-J15 was isolated from the jungle soil in Palenque National Park, Chiapas, Mexico. The strain formed pale brown, dry, tough, and buried colonies in the agar with no diffusible pigment in GAE (glucose–asparagine–yeast extract) medium. Scanning electron micrographs showed typical mycelium with long chains of smooth and oval-shaped spores (3–10 m). The strain grew in all of the International *Streptomyces* Project (ISP)’s media at 28–37 °C with a pH of 6–9 and 0–10% NaCl. *S. pakalii* ENCB-J15 assimilated diverse carbon as well as organic and inorganic nitrogen sources. The strain also exhibited significant inhibitory activity against the prodigiosin synthesis of *Serratia marcescens* and the inhibition of the formation and destruction of biofilms of ESKAPE strains of *Acinetobacter baumannii* and *Klebsiella pneumoniae*. The draft genome sequencing of ENCB-J15 revealed a 7.6 Mb genome with a high G + C content (71.6%), 6833 total genes, and 6746 genes encoding putative proteins. A total of 26 accessory clusters of proteins associated with carbon sources and amino acid catabolism, DNA modification, and the antibiotic biosynthetic process were annotated. The 16S rRNA gene phylogeny, core-proteome phylogenomic tree, and virtual genome fingerprints support that *S. pakalii* ENCB-J15 is a new species related to *Streptomyces badius* and *Streptomyces globisporus*. Similarly, its average nucleotide identity (ANI) (96.4%), average amino acid identity (AAI) (96.06%), and virtual DNA–DNA hybridization (67.3%) provide evidence to recognize it as a new species. Comparative genomics revealed that *S. pakalli* and its closest related species maintain a well-conserved genomic synteny. This work proposes *Streptomyces pakalii* sp. nov. as a novel species that expresses anti-biofilm and anti-quorum sensing activities.

## 1. Introduction

*Streptomyces* is a Gram-positive, mycelium-forming, sporulating bacteria with a high Guanine–Cytosine (G + C) content (57–75%). The genus belongs to the phylum Actinobacteria, the order Streptomycetales, and the family Streptomycetaceae. The life cycle of *Streptomyces* begins with the germination of their unigenomic dormant spores, the development of multigenomic filamentous hyphae or the mycelium substrate stage, and its subsequent autolysis for supporting morphological differentiations into aerial mycelium and sporulation [1,2].

The *Streptomyces* genus is abundant in terrestrial and marine habitats of the biosphere and may establish mainly mutualistic and pathogenic associations with plants, animals, and other microorganisms [3,4,5,6]. More than 700 *Streptomyces* species have been formally described [7], and 416 papers proposing new species have been deposited in the PubMed database since 1965. Although phylogenomic studies have re-clustered taxonomically many *Streptomyces* species [8], the bona fide genus maintains a comprehensive set of species. *Streptomyces* species excrete secondary metabolites of diverse chemical families with diverse biological activities and applications in industry and agriculture [9,10,11].

The increased isolation of microorganisms multi-resistant to antimicrobials has led to the search for new Actinobacteria strains capable of producing novel bioactive molecules [12]. As a result, the World Health Organization urgently called on public and private institutions to search for and develop new compounds to manage infections caused by antibiotic-resistant microorganisms (ARMs) [13]. Among the multiple targets for the control of bacterial infections is quorum sensing (QS) which is a cell-to-cell communication process that arranges the expression of virulence factors of many pathogenic bacteria [14]. Biofilm formation, mobility, secretion of extracellular enzymes, and pigment production are regulated by QS, among other cellular processes [15,16,17]. For this reason, the search for QS inhibitors has become relevant [18]. An example of QS mediated via N-acyl-L-homoserine lactone (AHL) is the synthesis of the red pigment prodigiosin from *Serratia marcescens*, which may be employed as a model to evaluate the activity of QS inhibitors [19].

Few findings have reported QS inhibitory activity in *Streptomyces*. For instance, *Streptomyces coeliflavus* produces three molecules (behenic acid, borrelidin, and 1H-pyrrole-2-carboxylic acid) that inhibit the production of pyocyanin and the expression of QS genes in *Pseudomonas aeruginosa* [20]. Additionally, *Streptomyces parvulus*, *Streptomyces albus*, and *Streptomyces violaceoruber* secrete actinomycin B, alnumycin D, and granaticin B which inhibit biofilm formation in ESKAPE bacteria and *Staphylococcus aureus* [21,22,23].

The availability of genome sequences and bioinformatics tools provides a new approach to taxonomy and contributes towards the discovery of expressed and cryptic biosynthetic gene clusters encoding bioactive molecules, which could significantly reduce the time and costs for the discovery of new drugs [24,25,26].

The current increase in antimicrobial resistance has led to the search for novel microorganisms with the potential for producing antibiotics. This work isolated a novel *Streptomyces* species capable of inhibiting biofilm formation and quorum-sensing activity in two ESKAPE bacteria and prodigiosin pigment biosynthesis in *S. marcescens*. This work aimed to analyze the phenotypic features, antimicrobial activities, and genomic features of the new species *Streptomyces pakalii*.

## 2. Materials and Methods

### 2.1. Sampling Area

The sampling area was in the Parque Nacional Palenque located in Chiapas, Mexico (17°30′33″ N, 91°58′56″ W). A total of 3 rhizospheric soil samples were obtained at a depth of 20 cm. The samples were transported to the laboratory at 5–10 °C.

### 2.2. Isolation and Purification of Actinobacteria

Serial decimal dilutions of the rhizospheric soil samples were performed using distilled water as a diluent in a final volume of 10 mL. The first 4 dilutions were spread onto glucose–asparagine–yeast extract (GAE) solid medium (20 g/L glucose, 1.0 g/L asparagine, 0.5 g/L yeast extract, 0.5 g/L K_2_HPO_4_, 0.01 g/L FeSO_4_, 0.5 g/L MgSO_4_, and 15.0 g/L bacteriological agar). After 5 days of incubation at 28 °C, typical colonies with morphology of actinobacteria (hard, dry, and buried colonies in agar) were reisolated in GAE medium by streak plate method until axenic cultures.

The strain ENCB-J15 was selected from among 15 different colonial morphotypes for further studies due to its remarkable capabilities to inhibit bacterial quorum sensing, biofilm formation, and biofilm destruction of two ESKAPE bacteria analyzed. Additionally, the strain was recognized early as a new species of *Streptomyces* by the preliminary 16S rRNA gene-based identification.

### 2.3. Growth and Maintenance of the ENCB-J15 Strain

Spores were obtained in GAE medium after 10 days of incubation at 28 °C, preserved with 30% glycerol in cryotubes at −70 °C, and lyophilized with 20% skim milk and 2% glycerol as cryoprotectants. The spores of the strain germinated, and mycelia grew on GAE solid medium at 28 °C for 5 days. Colonial and microscopic morphologies verified the purity of the strain. The strain ENCB-J15 was deposited in the collection of the Escuela Nacional de Ciencias Biológicas (ENCB) of Instituto Politécnico Nacional (IPN) and the Colección de Microorganismos of Centro Nacional de Recursos Genéticos, Instituto Nacional de Investigaciones Forestales, Agrícolas y Pecuarias, Mexico, which is a member of the World Federation of Culture Collection (https://wfcc.info/membership/memberlist, accessed on 15 May 2023).

### 2.4. Growth in Liquid Medium and Lyophilization of Supernatant

A suspension of spores from the strain ENCB-J15 was inoculated and incubated at 28 °C with orbital agitation at 150 rpm in an Erlenmeyer flask with GAE broth for 20 days. The microbial culture was centrifugated at 13,000 rpm for 15 min, and the supernatant was previously filtered through 0.45 and 0.22 μm sterile membranes for lyophilization. The products were rehydrated and concentrated at 10X (100 mg/mL) for further analysis.

### 2.5. Description of Microscopic Morphology by Scanning Electron Microscopy (SEM)

Petri dish microcultures of the strain ENCB-J15 were inoculated, grown in squares of GAE solid medium covered by a circular coverslip, and incubated at 28 °C for 10 days. The coverslip with a culture sample was placed on a 12-well microplate and fixed with 2% glutaraldehyde for 2 h, dehydrated with 10, 20, 30, 40, 50, 60, 70, 80, and 90% ethanol each for 10 min for each dissolution, and finally dehydrated with absolute alcohol for 20 min [27]. The sample was critically dried with CO_2_ using the Emitech K850 Critical Point Dryer to remove all moisture and subsequently mounted on aluminum sample holders with adhesive carbon tape. The sample was coated with a layer of gold at 20 mA for 2 min using Quorum Q15OR-ES equipment. Finally, the sample was observed in a Hitachi SU1510 SEM at 10–15 kW [28].

### 2.6. Phenotypic Characterization

The GAE solid medium was used as a basal medium to test the assimilation of carbon and nitrogen sources. The glucose was replaced by 1% saccharose, mannitol, lactose, glucose, fructose, galactose, xylose, starch, sorbitol, glycerol, maltose, galactose, or mannose. The asparagine was replaced by 0.5% NH_4_NO_3_, (NH_4_)_2_SO_4_, lysine, threonine, tyrosine, meat extract, meat peptone, casein peptone, or casamino acids. Additionally, the growth of the strain was evaluated in ISP 2, 3, 4, 5, and 7 media [29]. The growth of the strain was also evaluated in GAE solid medium adjusted to a pH range between 5 and 14, and NaCl concentrations ranged between 0 and 20%. The incubation temperatures were tested between 28 °C and 45 °C. All culture media were incubated at 28 °C for 7 days, except for the plates used for the temperature test.

### 2.7. Genomic DNA Extraction

The biomass of the strain ENCB-J15 was obtained from a culture of 5 days of growth in GAE solid medium. The DNA was extracted with the Soil Microbe DNA MiniPrep kit (Zymo Research, Irvine, CA, USA) following the manufacturer’s instructions. DNA integrity was verified by electrophoresis in 1% agarose gel, 1X TAE buffer, and staining with 0.5 µg/mL ethidium bromide solution. The DNA yield was 253 ng/μL.

### 2.8. Preliminary Molecular Identification Using 16S rRNA

The amplification of partial sequence belonging to 16S rRNA gene was performed via endpoint PCR using the universal primers 27F (5′-AGAGTTTGATCMTGGCTCAG-3′) and 1492R (5′-TACGGYTACCTTGTTACGACTT-3′) as previously reported [30]. The mastermix for 25 µL of PCR reaction contained 10X buffer, 25 mM of MgCl_2_, 10 mM of each dNTP, 10 pM of each primer, and 0.5 units of the Taq polymerase. The PCR reaction was carried out with the following thermal cycler conditions: one cycle of initial denaturalization at 94 °C for 5 min, 35 cycles of denaturalization at 94 °C for 1 min, alignment at 54 °C for 1 min, polymerization at 72 °C for 2 min, and a final polymerization at 72 °C for 10 min. Subsequently, 1% agarose gel electrophoresis was performed to corroborate the amplified product. The PCR products (approximately 1.4 kb) were purified with the Zymoclean^TM^ Gel DNA Recovery Kit (Zymo Research, Orange, CA, USA). The Sanger sequencing of the amplicons was performed at Macrogen^®^ laboratories in South Korea. BLAST compared the sequences with type strains of species of the genomic database bank NCBI and the List of Prokaryotic names with Standing in Nomenclature (LPSN), except when no type strains of the species were available.

The most closely related sequences were manually edited with the Seaview program [31]. A multiple alignment was performed by using CLUSTAL X [32]. The phylogenetic tree was constructed using the maximum likelihood method (ML) with the JC69 evolutionary model estimated with the MEGA v. 10 software [33]. The bootstrap method evaluated the tree topology’s robustness, computing 1000 repetitions [34].

### 2.9. Genome Sequencing and Annotation

The Whole Genome Sequencing (WGS) was performed at the Centre for Comparative Genomics and Evolutionary Bioinformatics (CGEB) at Dalhousie University, Canada, using the Illumina sequencing platform. The genomic libraries were prepared with the NexteraTM XT Library Preparation Kit from Illumina, and the sequencing reaction was carried out with the Illumina MiSeq equipment. The quality metrics from raw reads were analyzed using FastQC v. 0.11.8 (https://www.bioinformatics.babraham.ac.uk/projects/fastqc/, accessed on 25 August 2022). Raw reads were then trimmed using Trimmomatic v. 0.38 [35]. The de novo genome assembling was performed with the SPAdes v. 3.13.0 program [36]. The quality metrics of the assembly were determined with QUAST v. 5.0.2 [37]. The genome annotation was performed with Prokka v. 1.12 [38] and RAST v. 2.0 [39]. Prophage prediction was made with PHASTER [40]. Insertion elements (IE) were detected and annotated using ISEScan [41].

### 2.10. Phylogenomic Analyses

Phylogenomic reconstruction was performed using three methods: whole genome phylogeny by identifying the Virtual Genome Fingerprints (VGF), a core genome phylogeny (CGP), and a core-proteome-based phylogenomic analysis (CPBP). The VGF method was performed with the VAMPhyRE program [42]. CPBP was performed to obtain the clusters of orthologous groups of proteins (COGs) from the proteome of each organism with OrthoFinder v4.0 [43]. A maximum likelihood (ML) phylogenomic tree was constructed with the concatenated alignment of the COGs and LG + F + R7 model with IQ-TREE program [44]. The confidence level of the tree was estimated via ultrafast bootstrap with 1000 replicates. The CGP was built with 92 genes using the UBCG program [45]. The JC69 evolutionary test was used for phylogenomic reconstruction.

### 2.11. Determination of Average Nucleotide Identity (ANI), Average Amino Acid Identity (AAI), and In Silico Genome–Genome Hybridization (GGH)

ANI was determined by using FastANI [46], employing default parameters [47]. AAI was determined by using the AAI Calculator with default parameters (http://enve-omics.ce.gatech.edu/aai/, accessed on 18 November 2022). A matrix of identities for ANI and AAI was obtained, which was plotted with customized codes obtained with pheatmap version 1.0.8, written in R code (https://rdrr.io/cran/pheatmap/, accessed on 22 November 2022). In silico GGH was computed by the Genome-to-Genome Distance Calculator (GGDC 2.1) using the BLAST method [48]. The results were determined based on the recommended formula 2 (identities/HSP length). The threshold for deciding equal or different species was established based on species allocation criteria [47,48,49].

### 2.12. Comparative Genomic Analyses

The comparison analysis of functional annotation profiles was performed using Comparative_Genomics pipeline [50] (https://github.com/avera1988/Comparative_genomics, accessed on 10 December 2022) and the cdd2cog pipeline (https://github.com/aleimba/bac-genomics-scripts, accessed on 10 December 2022). COGs were compared and plotted with the customized scripts written in R code using the pheatmap package, version 1.0.8. Orthologous protein cluster comparisons were also evaluated with OrthoVenn2 Tool [51], establishing an E-value of 1 × 10^−5^ and an inflation value of 1.0.

Genome synteny analysis among the strain ENCB-J15 and the most related *Streptomyces* species (*Streptomyces badius* and *Streptomyces bacilliaris*) was performed with the progressive MAUVE tool included in the MAUVE software [52,53] using default parameters. Synteny comparisons were plotted with the genoPlot package of R [54].

### 2.13. Pathogenic Bacteria from the ESKAPE Group and Pigmented S. marcescens

The pathogenic strains used in this study were clinical isolates of *Acinetobacter baumannii* A15 and A22 and *Klebsiella pneumoniae* KpINP19, HP614, and KpINP8. They were donated by Dra. Graciela Castro-Escarpulli of Laboratorio de Investigación Clínica y Ambiental (ENCB, IPN). *S. marcescens* ENCB-MG01 belongs to the collection of the ENCB, IPN.

The strains were grown in Luria-Bertani (LB) solid medium (5 g/L yeast extract, 10 g/L casein peptone, and 5 g/L NaCl) at 37 °C for 24 h. After verifying their purity, the strains were preserved in 40% glycerol at −70 °C.

### 2.14. Inhibition of the Production of the Prodigiosin Pigment of S. marcescens by ENCB-J15 Supernatants

The production and inhibition of prodigiosin pigment of *S. marcescens* ENCB-MG01 were determined by mixing 1 mL of 1 × 10^8^ bacteria/mL with 1 mL of the 10X supernatant of the strain ENCB-J15 and incubated at 37 °C for 24 h. For the extraction of the pigment, the method described by Ramanathan et al. was used [55]. In brief, 100 mL of the bacterial growth obtained with or without the *Streptomyces* ENCB-J15 supernatant was placed in 1.5 mL microtubes and mixed with 100 µL of a solution of 2% HCL-absolute ethanol. The mixture was centrifuged at 5000 rpm for 5 min, and the supernatant was read at 535 nm for quantification in a Multiskan FC 357 Microplate Photometer (Thermo Scientific^TM,^, San Jose, CA, USA). The analyses obtained were evaluated via *t*-test and considered a significant difference as having a value of *p* < 0.0001.

### 2.15. Inhibition of the Formation and Destruction of Biofilms Formed by A. baumannii and K. pneumoniae

Multidrug-resistant clinical isolates of *A. baumannii* A15 and A21 and *K. pneumoniae* KpINP19, HP614, and KpINP8 were used. Each isolate was grown in 5 mL of medium LB at 37 °C for 24 h. The growth obtained was centrifuged at 13,000 rpm for 5 min, then the supernatant was removed, and the cellular button was washed with PBS 1x buffer 2 times. The inoculum was adjusted to 1 × 10^8^ bacteria/mL in RPMI 1640 medium (2 g/L D-glucose, 5 g/L phenol red, 6 g/L NaCl, 2 g/L NaHCO_3_, 1.512 g/L Na_2_HPO_4_, 0.4 g/L KCl, 0.1 g/L MgSO_4_, and 0.1 g/L CaNO_3_ at a pH of 7.2).

For the formation of bacterial biofilms, 200 μL of *A. baumannii* and *K. pneumoniae* suspensions and 100 μL of the actinobacteria supernatant ENCB-J15 were placed in polystyrene microplates of 96 sterile flat-bottomed wells and incubated for 24 h at 37 °C. For the destruction of preformed biofilms, the biofilms of each pathogenic strain were previously formed by placing 200 μL of bacterial inoculum in polystyrene microplates of 96 sterile flat-bottomed wells and incubating them at 37 °C for 24 h. After 24 h of incubation of biofilm formation, 100 μL of *Streptomyces* ENCB-J15 supernatant was placed and incubated for another 24 h. Biofilms were quantified with the methodology described by Christensen et al., 1985 [56].

The analyses obtained were evaluated by a two-way ANOVA analysis of variance with a significance level of *p* < 0.05.

## 3. Results and Discussion

### 3.1. Morphological and Phenotypic Characterization of Strain ENCB-J15

The strain ENCB-J15 cultured in GAE solid medium after 4 days at 28 °C developed buried colonies with a pale brown color, dry appearance, and tough consistency. The 7-day-old colonies acquired a whitish appearance on their surface due to aerial mycelium formation and spore maturation, but no diffusible pigments were observed (Figure 1A). SEM allowed us to observe typical mycelium and long chains of smooth and oval-shaped spores (Figure 1B,C). These morphological features are consistent with the typical features described for the genus *Streptomyces* [4,7].

The strain ENCB-J15 grew abundantly in the GAE solid medium and ISP media. However, its growth in the ISP3 medium was weak, and non-sporulation was observed after 5 days of incubation at 28 °C (Appendix A). Except for xylose, the strain grew in a GAE solid medium amended with different carbon sources tested, including monosaccharides, disaccharides, polysaccharides, and polyalcohol. Nonetheless, the growth of the strain in GAE + sorbitol was weak, and only vegetative mycelia were observed (Appendix A). The versatility of the strain ENCB-J15 reflects the high diversity but low availability of assimilable carbon sources in soil. Each carbon source displays a specific regulation of the biosynthesis of secondary metabolites determining the competence, defense, chemical communication, and survival of the microorganism in soil [57,58].

The characteristic pale brown pigment of the colonies was not observed in GAE amended with galactose and glycerol (Appendix A). All organic nitrogen sources that were used instead of asparagine in the GAE medium supported the growth and sporulation of the strain. However, inorganic nitrogen sources generated poor growth and sporulation (Appendix A). A nitrogen source is vital to the life cycle of *Streptomyces* as a component of proteins, nucleic acids, and many secondary metabolites [58].

The strain ENCB-J15 had a range of temperatures and pH growth typical of most Streptomycetaceae species (Appendix A) [7,59]. At the maximum growth temperature of 37 °C, the strain ENCB-J15 synthesized a non-diffusible melanin-like black pigment. Bacterial pigments are often synthesized in non-optimal or stress culture conditions, with various temperatures and pH values. Melanins generally protect microorganisms from environmental stress conditions, such as ultraviolet radiation and oxidative stress by heavy metals [60]. Several *Streptomyces* species secrete tyrosinases that produce soluble melanins in media [61], but the black pigment of the strain ENCB-J15 is attached to biomass.

Finally, the strain ENCB-J15 tolerated NaCl concentrations below 10% (Appendix A). Halotolerance to high concentrations of NaCl is commonly observed in Actinomycetes, such as *Saccharopolyspora ghardaiensis*, which is capable of optimal growth in concentrations between 15 and 25% NaCl [62]. However, most Actinobacteria species, including the *Streptomyces* genus, tolerate only 1% NaCl [63].

### 3.2. Phylogenetic, Phylogenomic, and Pairwise Comparison

The strain ENCB-J15 was located in an independent branch of the 16S rRNA phylogenetic tree, but no clear differentiation among related species was observed (Figure 2). The percentage of nucleotide similarity between the strain ENCB-J15 and its closest phylogenetic relatives ranged from 93.5 to 94.1% (Table 1). The widely accepted cutoff for delimiting new bacterial species using 16S rRNA gene sequences is between 97 and 98.65% [64,65,66,67,68,69,70]. Under this criterion, the ENCB-J15 strain could be considered a new species. Currently, it is assumed that the 16S rRNA gene sequence is not sufficient for species-level identification, differentiation between related species, and the definition of new bacteria species [68,71,72,73], and the *Streptomyces* spp. are no exception [8,74,75,76]. The identification of *Streptomyces* species based on a multilocus sequence analysis (MLSA) of *atpD*, *recA, trpB, rpoB*, and *gyrB* genes is a more robust molecular tool for species-level identification [76,77,78,79].

Currently, whole-genome sequencing methods and bioinformatic analyses of the G + C content, ANI, AAI, in silico GGH percentages, and phylogenomic reconstructions make it possible to derive useful information about taxonomy and phylogeny. The G + C content measured by chemical analyses allowed a maximum variation within bacterial species of 3–5 mol% [64,80]; meanwhile, the G + C content calculated from the genome sequence proposes a maximum variation value into bacterial species of 1% [48]. The ENCB-J15 strain has a G + C content of 71.63%, which is in the range of the *Streptomyces* genus (69.7–74.5%) [8].

In addition, more and more often, ANI, AAI, and in silico GGH percentages are used as essential criteria to define cutoff values of species and assign new species. The following cutoff values of ANI (95–96%), AAI (<95%), and in silico GGH (<70%) are frequently used to define species limits in bacteria [48,49,81]. Nonetheless, the ANI cutoff value of 96.5% is also accepted since this value offers better resolution for bacterial species delimitation [68,82]. The ANI cutoff value of ≥96.5% has been proposed for delineating species belonging to *Streptomyces* and members from Actinobacteria, such as *Salinospora* [68,83].

The features of the ENCB-J15 genome are outlined in Table 2. According to ANI and AAI, *S. badius* SP6C4 and *S. globisporus* TFH56 type species displayed very high values for ANI and AAI between them (99.99 and 99.98%, respectively), but a lower ANI of 96.43% and AAI of 96.06–96.01% with the ENCB-J15 strain. Both indexes were even lower among the strain ENCB-J15 and other *Streptomyces* species (Figure 3). An in silico GGH analysis between the ENCB-J15 strain and its closest relatives *S. badius* and *S. globisporus* showed a value of 67.3% (Table 1). Recently, in silico GGH has been recognized as a valuable criterion of relatedness and has replaced the experimentally complex and variable DNA–DNA hybridization assays [48,84].

The sequencing of whole genomes and the phylogenomic approach have become valuable tools to define and recognize new bacteria species for science [8,85,86,87,88]. CGP, CPBP, and VGF were used to perform the phylogenomic approaches of the strain ENCB-J15, but the most robust maximum likelihood phylogenetic tree was obtained based on CPBP. ENCB-J15 was closely clustered in this tree with *S. badius* and *S. globisporus*, but *S. filamentosus* and *S. parvus* were more distant (Figure 4). In all the phylogenomic trees, *S. badius* and *S. globisporus* formed a clade and shared a close common ancestor; meanwhile, the speciation event of the strain ENCB-J15 was earlier (Figure 4 and Appendix A). Currently, phylogenomic reconstruction is the most robust tool for species classification in bacteria in general and the Actinobacteria phylum in particular [8,86]. Thus, the topology of both phylogenomic trees is consistent with the proposal of ENCB-J15 as a new species. There are non-official criteria for categorizing bacterial species based on phylogenomic metrics. A CGP based on at least 100 single core genes could resolve taxonomical discrepancies among closely related strains [85,89]. Even for the *Streptomyces* genus, CPBP based on approximately 700 COGs allows for clusters and defines the species into a conscious and well-resolved phylogenetic tree [68]. Furthermore, confidence trees with well-supported nodes with bootstrap values ≥70% have been suggested [85]. Although the phylogenomic metrics for bacterial species designation are in the normalization process, this work fulfills the species identification of ENCB-15 based on phylogenomic reconstruction with the metrics described above.

*S. badius* and *S. globisporus,* formally recognized as independent species, have genomic content, indexes, and percentages that suggest they are two strains of the same species. Nonetheless, the ENCB-J15 strain has G + C content, ANI, AAI, in silico GGH percentages, and phylogenomic trees that are consistent with the proposed metrics to assign a strain as a new species which we suggest can be named *Streptomyces pakalii* sp. nov.

### 3.3. Analysis of Genomic Features among Streptomyces pakalii sp. nov. and Closely Related Species

The genome of *S. pakalii* sp. nov. has a size of 7.6 Mb, a 71% G + C content, 6833 genes, and 6746 proteins annotated (Figure 5 and Table 2). Additionally, four ribosomal genes, 82 tRNA genes, four sequences corresponding to putative prophages, and five insertion elements (IE) were detected (Table 2 and Appendix A). *S. pakalii* sp. nov. showed a similar genome size and higher protein content than *S. badius* and *S. globisporus*, its most proximal relatives. Nevertheless, these species present more rRNA and tRNA gene copies than *S. pakalii* sp. nov. ENCB-J15 [90]. The results suggest that *S. pakalii* sp. nov. probably suffered encoding-gene expansion into its genome, as well as a putative ribosomal and tRNA depuration. The encoding-gene gain is a common phenomenon in the genus *Streptomyces* that helps it to adapt to the environment [91]. Although plasmids were not evidenced, other mobile elements were detected, such as prophages or insertion elements (Table 2).

The synteny analysis of the entire genome reflected a similar genome structure among *S. pakalii* sp. nov., *S. badius*, and *S. bacilliaris*. However, *S. pakalii* sp. nov. displayed events of reordering and gene insertions (Figure 6). As previously observed, the chromosome structure in the species analyzed was highly conserved at the central region generally associated with the core genome. However, the insertions were enriched at the terminal regions of the chromosome, a region corresponding to the accessory genome that includes metabolite biosynthetic genes [92].

### 3.4. Comparative Analysis of Functional Annotation

*S. pakalii* sp. nov. ENCB-J15 showed 6337 proteins with known functions and 409 with non-determined functions. Most of the annotated genes of *S. pakalii* sp. nov. were assigned roles in transcription, amino acids, carbohydrate transport, and metabolism (684, 458, and 456 genes, respectively) (Appendix A). COGs with large repertories were related to lipid transport and metabolism, energy production and conversion, translation and biogenesis, cell wall development or biogenesis, signal translation mechanisms, and inorganic ion and transport. All genomes showed an extensive repertory of COGs related to secondary metabolite biosynthesis (SMB) with a content of 218–277 COGs (Figure 7A). The analysis with OrthoVenn indicated that 4812 core COGs are shared among the *Streptomyces* species included in this analysis (Figure 7B). *S. pakalii* sp. nov. exhibited 26 accessory COGs, mainly involved in carbon source and amino acid catabolism, DNA modification, and the antibiotic biosynthetic process (Figure 7B and Appendix A). Core genomes are frequently involved in primary metabolism and DNA processing functions, whereas genes associated with SMB are increased in the *Streptomyces* pangenome [91,93]. This suggests that acquiring accessory genes is a central phenomenon associated with speciation in *Streptomyces* [94]. The complexity of forest soil microhabitats may maintain selective pressures that explain the vast repertoire of accessory genes and the phenotypic versatility of *S. pakalii* sp. nov.

### 3.5. Inhibition of Prodigiosin Biosynthesis of S. marcescens by S. pakalii sp. nov. ENCB-J15 Supernatants

The expression of diverse bacterial virulence factors, such as biofilm formation in pathogenic bacteria and the biosynthesis of pigments such as prodigiosin in *S. marcescens,* depends on QS [15]. Therefore, QS has been pointed out as a possible target for controlling pathogenic bacterial infections [95,96]. In this work, the supernatant of *S. pakalii* sp. nov. was inhibited until 50% of the biosynthesis of prodigiosin of *S. marcescens, which* was achieved with no decrease in bacterial growth (Figure 8). Nonetheless, more research should be conducted to determine if the target of the inhibitory activity is at the level of pigment biosynthesis or QS. Hence, assays of the inhibition of prodigiosin biosynthesis are a tool to recognize potential QS inhibitors.

### 3.6. Inhibition of the Formation and Destruction of Biofilms Formed by A. baumannii and K. pneumoniae

One of the most important virulence factors today is the formation of biofilms, which play a critical role in human pathogenic bacteria such as *A. baumannii* and *K. pneumoniae*, since they act as a structure that increases resistance to different agents, such as antimicrobial compounds, detergents, and disinfectants, among others, and prevents their action, causing the microorganisms that form these biofilms to become resistant [55]. The supernatant of *Streptomyces pakalii* sp. nov. displayed anti-biofilm activity because it was able to inhibit more than 50% of the formation of these (Figure 9A), as well as destroy the biofilms already formed (Figure 9B) by the ESKAPE isolates of *K. pneumoniae* and *A. baumannii*. It showed activity similar to that reported by Sangkanu et al. in 2017 [97], whose results showed a decrease in the formation and destruction of biofilms formed by *Staphylococcus epidermidis* ATCC35984 in the presence of the supernatants of different actinobacteria isolated from a mangrove swamp in Thailand. These results suggest that a molecule, hitherto unknown, in this supernatant may be used as an anti-biofilm agent.

### 3.7. Description of Streptomyces pakalii sp. nov.

*Streptomyces pakalii* (Etimology: pak.al’i.i N.L. gen. masc. n. pakalii; the name is derived from the Mayan King K’inich Janaab Pakal, who was buried in a Mayan pyramid) is a Gram-positive filamentous bacterium that shows the formation of smooth spore chains. The front and back of the colony have a pale brown color without the presence of diffusible pigments in culture media. Its growth was observed at temperatures of 28–37 °C with the optimal growth at a temperature of 28 °C. When it grows at 37 °C, it produces a non-diffusible melanin-like pigment in GAE medium. The strain grew in NaCl concentrations up to 10% and pH values of 5–13. It uses glucose, starch, maltose, lactose, sucrose, mannitol, glycerol, sorbitol, fructose, galactose, and mannose but cannot use xylose as a carbon source. Alanine, proline, serine, glutamine, phenylalanine, aspartic acid, asparagine, casamino acids, casein peptone, malt extract, meat extract, meat peptone, and yeast extract were used as nitrogen sources. Nonetheless, NH_4_NO_3_ and (NH_4_)_2_SO_4_ were mediocre nitrogen sources. The strain grew properly in ISP-2, ISP-3, ISP-4, ISP-5, and ISP-7 media. The evidence provided by its phenotypic traits, G + C content, ANI, AAI, in silico GGH percentages, phylogenomic reconstructions, and comparative genomics supports our proposal that *S. pakalii* ENCB-J15 should be considered as a new species within the *Streptomyces* genus. Regarding its biological activity, it is suggested that the compounds produced by this strain may have anti-QS activity, specifically inhibiting and destroying biofilms of ESKAPE strains of *A. baumannii* and *K. pneumoniae* and inhibiting the production of the prodigiosin pigment of *S. marcescens*.

## 4. Conclusions

Our evidence suggests that increasing genome size, associated with gene gain, probably contributed to the speciation of *Streptomyces pakalii* sp. nov. Our functional analysis could explain certain phenotypic traits observed in our study strain since it could assimilate all the carbon and nitrogen sources tested in this work. In future studies, it is necessary to perform a more robust functional analysis of the genome to understand how the slight genome revolution modified the metabolic features of our strain of study.

## Figures and Tables

**Figure 1 microorganisms-11-02551-f001:**
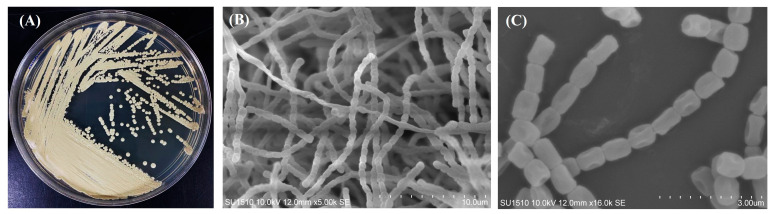
Morphology of *Streptomyces pakalii* sp. nov. ENCB-J15. (**A**) Colonial morphology in glucose–asparagine–yeast extract solid medium (GAE); (**B**) Abundant chains of spores and aerial mycelium observed by scanning electronic microscopy (SEM); (**C**) Smooth-looking spore chains observed by SEM (Hitachi SU1510).

**Figure 2 microorganisms-11-02551-f002:**
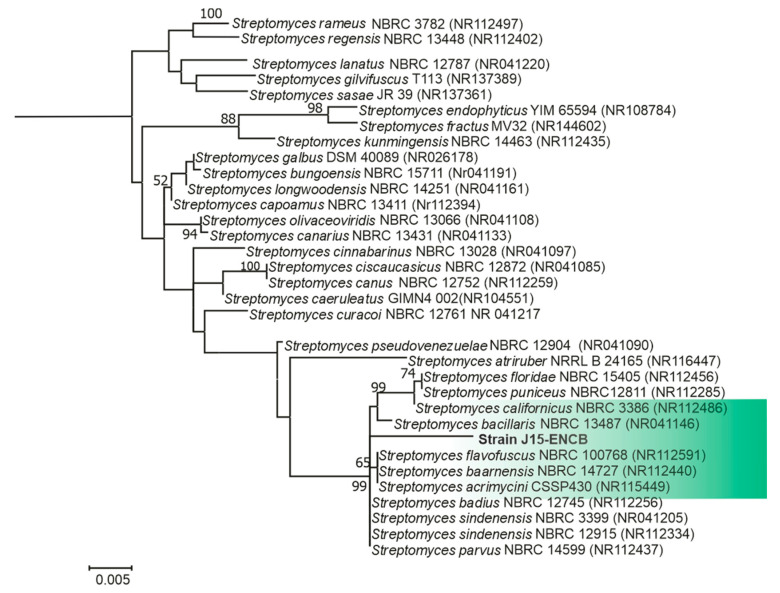
Maximum likelihood phylogenetic tree of strain ENCB-J15 and closely related species based on 16S rRNA gene sequences (1500 nt). *Streptomyces pakalii* sp. nov. ENCB-J15 is highlighted in the green box. Numbers over the branches are the bootstrap values >50% computed by 1000 replicates. The scale bar represents the number of substitutions per base.

**Figure 3 microorganisms-11-02551-f003:**
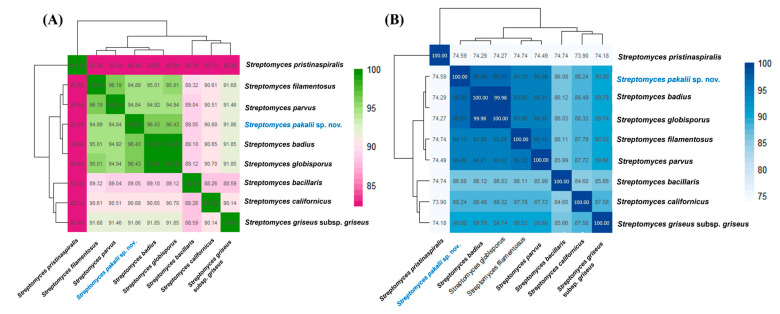
Heatmaps of Average Nucleotide Identity (ANI) (**A**) and Average Amino Acid Identity (AAI) (**B**) calculated in pairwise comparisons of related *Streptomyces* species. The color gradient bar at the *x*-axis represents the ANI and AAI values. *Streptomyces pakalii* sp. nov. ENCB-J15 is highlighted in blue.

**Figure 4 microorganisms-11-02551-f004:**
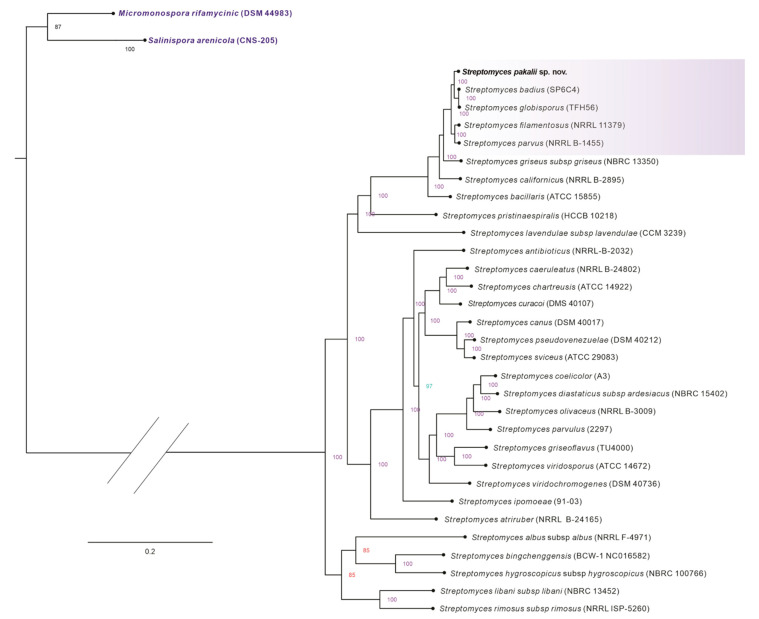
Maximum likelihood phylogenomic tree of *Streptomyces pakalii* sp. nov. ENCB-J15 and other related species of the genus *Streptomyces*. The phylogeny was based on the core-proteome of 1218 Clusters of Orthologous Groups of Proteins (COGs) and reconstructed with Orthofinder and IQ-TREE. Numbers over the nodes represent the ultrafast bootstrap values >50% computed by 1000 replicates. The scale bar represents the number of substitutions per base. *Streptomyces pakalii* sp. nov. ENCB-J15 is highlighted in the purple box.

**Figure 5 microorganisms-11-02551-f005:**
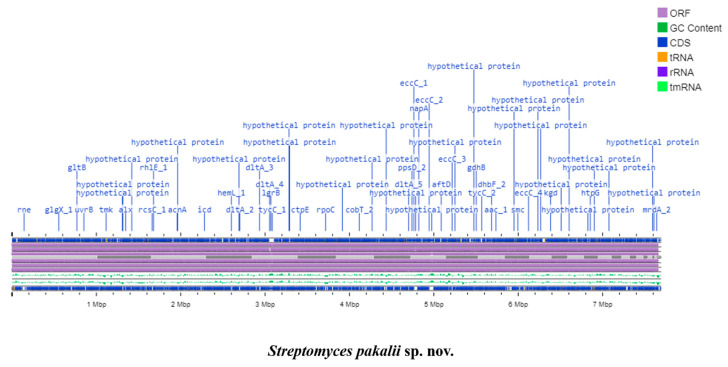
Linear genome map of *Streptomyces pakalii* sp. nov. ENCB-15. Color code from the center out: genome size (first line), coding sequences (CDS) (blue), G + C content (green), Open Reading Frames (ORFs) (purple), and genome contigs (grayscale).

**Figure 6 microorganisms-11-02551-f006:**
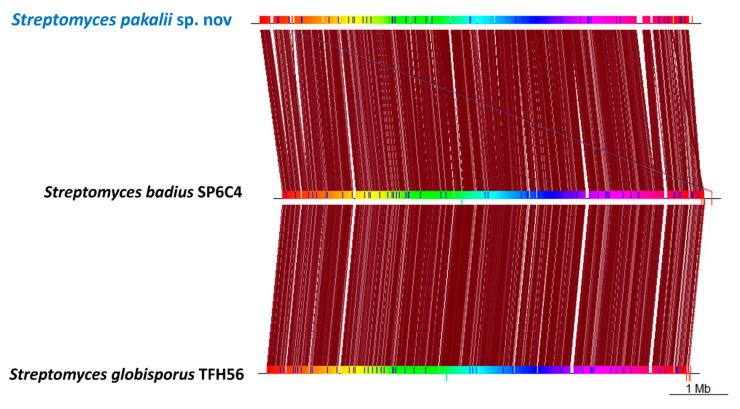
Genomic synteny and comparison among *Streptomyces pakalii* sp. nov. ENCB-J15 and related species. Color code: conserved regions (rainbow shades), variable regions (white gaps), conservation of the direction or position of the regions (red lines), and inversions or translocations (blue lines).

**Figure 7 microorganisms-11-02551-f007:**
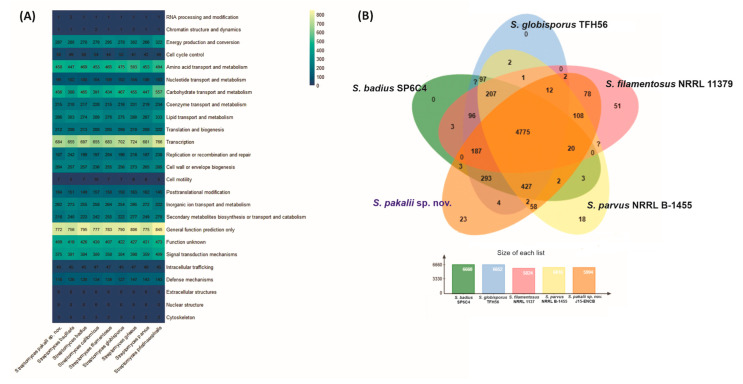
Predicted protein content comparison among *S. pakalii* sp. nov. ENCB-J15 and closely related species. (**A**) Heatmap of clusters of orthologous groups of proteins (COGs)’ abundance sorted into different functional categories. The gradient color bar represents the relative content of COGs. Numbers in the boxes represent the absolute content of COGs. Names on the *x*-axis represent the organisms analyzed. The functional categorization is represented on the *y*-axis. (**B**) Venn diagram of shared and accessory COGs among *Streptomyces pakalii* sp. nov. ENCB-J15 and close species. Numbers in the diagram represent the shared and accessory content of COGs. The bar plot represents the overall content of COGs in each *Streptomyces* species included in the analysis.

**Figure 8 microorganisms-11-02551-f008:**
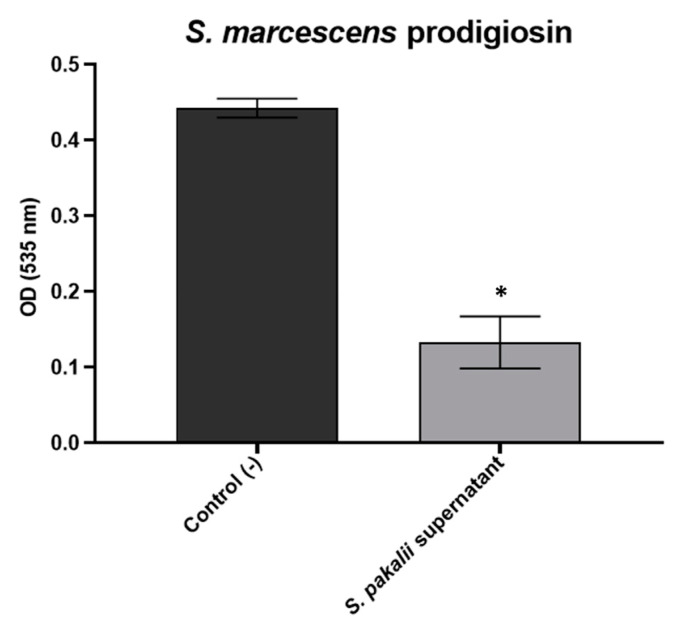
Inhibition of the production of the prodigiosin pigment of *S. marcescens* by *Streptom-ces pakalii* sp. nov. ENCB-J15 supernatants. The symbol on the bar (*) represents a statistically significant difference by *t*-tests. *p* < 0.0001.

**Figure 9 microorganisms-11-02551-f009:**
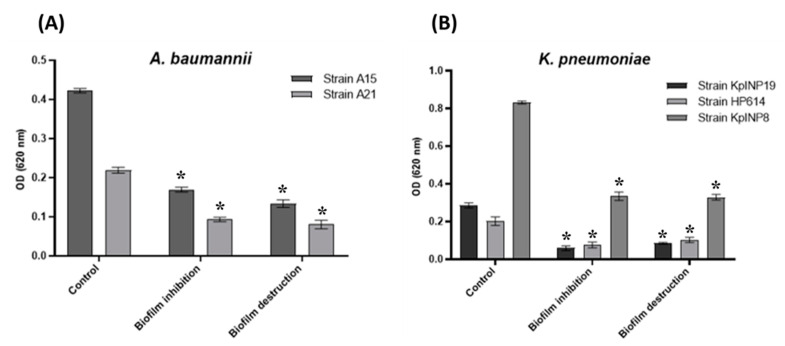
Inhibition and destruction of ESKAPE bacteria biofilms by *Streptomyces pakalii* sp. nov. ENCB-J15 supernatants. (**A**) Inhibition and destruction of *A. baumannii* (A15 and A21) biofilms; (**B**) Inhibition and destruction of *K. pneumoniae* (KpINP19, HP614, and KpINP8) biofilms. The symbol on the bar (*) represents a statistically significant difference by a two-way ANOVA analysis. *p* < 0.05.

**Table 1 microorganisms-11-02551-t001:** Comparison of nucleotide similitude percentages for 16S rRNA gene, genomic G + C content, and in silico genome–genome hybridization (GGH) of *Streptomyces pakalii* sp. nov ENCB-J15 with related species.

	*Streptomyces pakalii* sp. nov. ENCB-J15		
*Streptomyces* Species	Nucleotidic Similitude Percentages of 16S rRNA Gene (%)	Genomic G + C Content (%)	In Silico GGH Percentages (%)
*Streptomyces badius* SP6C4	94.1	71.60	67.3
*Streptomyces globisporus* TFH56	93.5	71.54	67.3
*Streptomyces parvus* NRRL B-1455	94.8	71.60	57.2
*Streptomyces filamentosus* NRRL 11379	94.2	71.30	56.3
*Streptomyces griseus* subsp. *Griseus* NBRC 13350	94.4	72.20	41.0
*Streptomyces californicus* NRRL B-2895	94.4	72.50	38.3
*Streptomyces bacillaris* ATCC 15855	93.8	71.5	33.1
*Streptomyces pristinaspiralis* HCCB 10218	93.9	71.50	23.3

**Table 2 microorganisms-11-02551-t002:** Genome features of *Streptomyces pakalii* sp. nov. ENCB-J15.

Feature	Chromosome Characteristics
Genome topology	Linear
Chromosome size (pb)	7,659,259
# Contigs	36
N50	461,892
L50	6
G + C content (%)	71.63
Total of genes	6833
Protein coding-genes	6746
Genes assigned to COG	5562
rRNA genes	4
tRNA	82
Prophages	4
Insertion elements	5
Plasmids	No detected

## Data Availability

The Whole Genome Shotgun project has been deposited at DDBJ/ENA/GenBank under the accession number JARWAF000000000. The version described in this paper is number JARWAF010000000. The accession number for the sequence corresponding to the 16S rRNA gene deposited in the GenBank is OQ982082.1.

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
