# Peer review of "Phenotypic and Genomic Characterization of Streptomyces pakalii sp. nov., a Novel Species with Anti-Biofilm and Anti-Quorum Sensing Activity in ESKAPE Bacteria"

_microorganisms, 2023, doi:10.3390/microorganisms11102551_

Round 1
Reviewer 1 Report
The study introduces a new species of Streptomycetaceae family, named Streptomyces pakalii sp. nov. The strain ENCB-J15 was discovered in the jungle soil of Palenque National Park, Mexico. The strain showed typical mycelium with long chains of smooth and oval-shaped spores and grew in all The International Streptomyces Project (ISP) media. S. pakalii ENCB-J15 had a high G+C content and 6,833 total genes, with 6,746 genes encoding putative proteins. The study also revealed that the new species exhibited significant inhibitory activity against prodigiosin synthesis of Ser ratia marcescens and inhibition of the formation and destruction of biofilms of ESKAPE strains of Acinetobacter baumannii and Klebsiella pneumoniae.
The study used a complete set of methods to characterize in detail the new species. The methods were adequately described, following rigorous standards, and the results were clearly presented. Other than minor text editing, the manuscript can be accepted for publication.
I congratulate the authors for the discovery and characterization of this novel species.
Just minor text editing is necessary.
Author Response
Thank you very much for the comments on our work's manuscript. A new revision of the English style was carried out.

Reviewer 2 Report
my comments
1- This abstract requires improvement by adding a brief introduction and conclusion to enhance its clarity. also the results are represented well in it.
2- The introduction section lacks the goal of the work at the end of the introduction.
3. in line 55 rewrite Is to is
4. In the introduction, it is important to discuss the role of Streptomyces or actinobacteria in biofilm inhibition and quorum sensing, drawing upon previous research findings.
5. In material and methods, you should include a subtitle about soil sample collection that includes number of samples, the location axis, city, country, and method of collection.
6. In material and methods, Under the subtitle, isolation and purification of actinobacteria, you should write the method of isolation in detail with appropriate references, as well as the name of culture media, soil diluted or not, and the method of purification.
7. How many isolated actinobacteria did you obtain from the soil samples?
8. What criteria did you use to choose the target isolate? The ENCB-J15 strain? Write this information in the manuscript.
9. In line 82, which GAE broth, all names should be written full name the first time.
10. You should mention the number of isolated actinobacteria in the results section, do primary screening for their ability to inhbition biofilm and QS, and then select target strain.
11. In material and methods, you should include a subtitle about the tested pathogenic bacteria, including where you got it and how it grew.
12. In line 182, you wrote which previous reported method, please write the name of this method in the details.
13. In line 187, you should determine exactly 2 or 3 pathogenic bacteria used in the study.
14. Rewrite all chemical format symbols and number superscripts using the corrected method.
15. The method used in lines 177 to 202 is unclear; clarify.
16. Figure 9: What are the strains mentioned in the figures that you did not mention previously?
17. Table 1 for the isolate's 16s rRNA or total genome.
18. The PCR method is unclear, describe it in detail, including the types of primers.
Minor editing of English language required
Round 2
Reviewer 2 Report
Accept in present form